# Climate change, geography and trade agreements: A perspective of Asian bilateral trade

**Qianxu Liang[1], Lin Shao[1]\*, Zahid Hussain[2]\*, Yufang Chao[1], Haiying Liu[1], Chaonan Wang[1]**

**1** Department of Economics and Management, Qilu University of Technology (Shandong Academy of Sciences), Jinan, PR China, **2** Business School, Faculty of Economics, Liaoning University, Shenyang, P.R. China

\* linshao@qlu.edu.cn (LS); zahiduibe@yahoo.com (ZH)

**Data availability statement:** This data is not collected from third party. Therefore, other

## Abstract

This study investigates the simultaneous effects of geographic factors, trade agreements, and climate change on bilateral exports in Asian countries. We estimate the correlation with bilateral exports by utilizing a panel data set from 2000 to 2020, employing various econometric techniques, particularly the structural gravity model. Therefore, this study aims to examine the simultaneous or complementary impact of influencing factors on exports and link them with the gross domestic product. Findings demonstrate that geographic factors are crucial for determining bilateral exports in terms of increasing trends. Furthermore, geography plays a crucial role in enhancing the magnitude and probability of bilateral exports between trading partner countries. Moreover, bilateral exports have declined because of the simultaneous impact of geographic factors, climate change, and economic size. Thus, geographic factors and economic size affect marginal exports to varying degrees. This study suggests that the simultaneous increase in economic size, trade agreements, and geographic factors can enhance the bilateral export level and its probability at an above-average rate between trading partner countries.

## Introduction

Exports always play a vital role in economic growth and development for each economy; however, they are affected by several factors, especially natural and man-made geographic factors [1, 2]. Therefore, geography is categorised into two aspects: first, natural geography, which comprises the physical fundamentals of nature, such as mountains, seashores, and natural resources. Second, man-made geography comprises the fundamental's spot of diverse economic actions [3]. However, while challenges and facilities are created by geographic factors, they remarkably impact firms' decisions regarding exports, infrastructure, and market access [4–6]. Distance is a crucial factor for trade, which significantly increases international trade costs via the transportation system. More analytically, the distance is detected via geographic coordinates, e.g., latitude and longitude. Thus, the differentiation in coordinate points stimulates the transport infrastructure level (which is a determinant of trade costs) [7,8].

authors can directly collect the data from given sources (websites). Data is publicly available at different websites. It is stated that the data is unable to upload on any link or system after several attempts. So, it is advised that the data can be downloaded from recommended websites. The reason is that there are 44436 observations, which are extremely large. Besides, data is not only country level but also bilateral (country 46* country*46*year21=44436). The panel dataset covers the 46 Asian countries that collect information on variables period from 2000 to 2020 through several international organizations (e.g., the World Bank, CEPII, and the United Nations Conference on Trade and Development (UNCTAD). The bilateral exports data is collected by UNCTAD website https://unctad.org. The data on tariff, population and gross domestic product variables is available on the website of World Development Indicators (WDI) https://databank.worldbank.org/source/world-development-indicators. Therefore, the data on variables namely distance, access to Ocean and common border is available on the website of CEPII https://www.cepii.fr/cepii/en/bdd_modele/bdd_modele.asp. In addition, the trade agreement (PTA)'s data are collected from WTO website, https://www.wto.org. PTA dummy variable refers to one if exporter and importer listed were in any one of PTA's listed either the WTO website https://www.wto.org. Besides, the data on CO2 emissions is available from the World Input-Output Database (WIOD) website https://www.rug.nl/ggdc/valuechain/wiod/.

**Funding:** The author(s) received no specific funding for this work.

**Competing interests:** The authors have declared that no competing interests exist.

In addition, climate change indirectly affects exports through the weather changes because firms export the weather-oriented goods. Climate change is varied because of the high level of $CO_2$ emissions [9,10]. Besides, $CO_2$ emission is an important element in climate change, and can directly or indirectly affect the environment, including the natural geographic factors such as temperature, mountains, and seashores [11]. Man-made geographic factors also substantially affect export-related activities through firms' strategies (e.g., increasing return to scale, transportation costs, and trade costs) [12]. However, the influence of these factors is geographically varied. Countries are rich with natural and man-made resources, may gain from trade, whereas countries with poor geographic factors cannot achieve export targets. Nevertheless, the literature shows that traditional theories are derived based on the above geographic concepts instead of giving great attention to empirical ways that are directly or indirectly related to exports.

Notably, some man-made factors particularly trade agreements can have substantial influence on bilateral exports, which are signed based on several reasons such as economic size, geographic factors (i.e., distance, common border), bilateral, multilateral political and cultural similarities and so on. Therefore, countries are experiencing exceptional growth of trade agreements. Recently, more than three hundred preferential trade agreements (PTAs) have been implemented globally. The initial trade agreement-related activities were solely regional, and lifted out the sanctions with restriction of article 24 of GATT, however, afterward PTAs, connecting the countries or groups of countries from distinctive regions.

These trade agreements could be bilateral or plurilateral (common markets, economic and custom unions). They enlarged the scope of negotiations to areas which out of WTO mandate, and become favorable policy mechanisms. In addition, trade agreement arrangements operate debate for trade-related policy issues. The foundation, span, and reporting of these trade agreement-related arrangements are diverse from each other, and exhibit the extensive heterogeneity. Interestingly, PTAs are classified into two aspects: shallow and deep integration. So, the shallow integration indicates reduction or elimination of trade barriers, while hidden incorporation refers to coordination of domestic policies and concealed border measures [13,14].

According to World trade organization (WTO), [15] the depth and coverage of PTAs management diverges from regional trade agreement (RTA) to another. Currently RTAs have tendency to go extremely afar tariff-cutting exercise but not utterly those associating with most developed economies. They offer progressively multifaceted policies to govern intra-trade with respect to custom management, protection and standards including preferential monitoring charter for common trade facilities. Furthermore, a record high-level RTAs go beyond conventional trade policy mechanisms involve the reginal rules on competition, labor and investment.

This study concentrates on the fundamental question of whether PTAs contribute to bilateral exports or adversely affect them, as previously assumed by author[16]. The prevailing view suggests that PTAs add value to the development of international trade based on traditional trade agreements while simultaneously operating at the multilateral level [17]. Thus, it is notable that negotiations are required to reinvestigate despite the border measures. The trade agreements are complicated, and their content may change drastically from agreement to agreement [18].

Besides, former literature affianced investigation to present distinction clauses in PTAs such as WTO+ and WTOX by introducing [19–22] in an article. The distinction WTO+ refers to negotiable areas which fall under the WTO mandate such as customs administration, tariff barriers, export and import restrictions and technical barriers to trade etc., whereas WTOx distinction indicates the independent of WTO mandate, e.g., mobility of the labor,

competition policy and environmental standards. Some studies, e.g., [23] and [24] analyzed the impact of WTO$^+$ and WTO$^x$ on trade flows separately based on distinction structure. Their findings are contradicted to each other in relationship between trade and WTO$^x$ clauses. However, a scanty analysis was considered WTO$^x$ clauses as a substantial aspect evaluated by numerous studies. Each area of WTOx remained to be unidentified as a certain contribution. Thus, this paper emphasizes on clauses in WTOx category by presenting the labor mobility, competition policy, capital mobility and environmental standards [25].

The present study aims to analyze the simultaneous influence of geographic factors, climate change, and trade agreements on Asian bilateral exports and investigate how geographic factors significantly impact exports. Estimating the influence of geographic factors on exports gives a clear picture of the amount of investment needed to improve man-made or natural resources. This study adds to the literature by exploring geographic factors that affect bilateral exports. Related literature has failed to incorporate the estimation of the simultaneous impact on bilateral exports. Simultaneous impact identifies whether natural or man-made geographic factors (e.g., physical distance, ocean access, and shared common border) separately affect bilateral exports. Moreover, a contribution includes the one-belt, one-road regions, especially Asian countries, which are the most diverse geographically. In addition, this paper adds value by investigating the role of trade agreements in bilateral exports in depth. The clauses of the WTOx distinction provide pathways to ensure the environmental standards, safeguards, and quality of exported commodities. Thus, it is essential to investigate the effect on bilateral exports. Lastly, this study links climate change ($CO_2$) emissions and gross domestic product (GDP) with bilateral exports to estimate substitutability and simultaneous impact.

We use panel data at the bilateral level for 46 Asian countries and utilize export equations, including control variables. The heterogeneity of a country's characteristics supports the adoption of controlling approaches that are correlated or uncorrelated with explanatory variables and error terms in this model. Moreover, this paper deals with potential bias due to heterogeneity and model selection preconceptions by using the [26]. The outcomes illustrate that developments in geographical factors can expand the export level between the countries. Improving the quality and quantity of natural geographic factors enhances bilateral exports. Besides, the results indicate that both geographic factors complement each other when determining the bilateral export level between trading partners. We attempt to reverse the potential reverse causality of exports to geography, climate change, and trade agreements through changes in the measurement scale, which cannot affect the export volume.

The discussions in the different sections are as follows: Related studies are explained in the previous work section. The conceptual framework, estimation model, and econometric tools are presented in the methodology section. The experiment and discussions are reported in the discussion section. The conclusions section concludes with recommendations and future research directions.

## Previous work

Several studies have explained the effects of geographic factors, $CO_2$ emissions, and trade agreements on international trade based on different models. However, our concern is estimating the simultaneous impact on bilateral exports by considering geographic factors.

### Nexus between geographic factors and trade

This strand of literature explains the correlation between geographic factors and trade. Several studies, e.g., [27] argued that climate change have remarkable impact on trade approximately 54% and 56% in high-income and upper-medium income countries respectively. Another

study, e.g., [28,29] find that complementarity and comparative advantage drastically stimulate the bilateral exports between China and Vietnam in agricultural sector based on geographical factors such as location, seashore and border sharing over time. However, partner countries have win-win situation by trading agri-based products under export-strategies. The author [1] discussed that geographic factors such as climate change potential and ocean access have a simultaneous effect on international trade. Precisely, climate change potential affects international trade through access to the oceans, indicating that the oceans are encompassed by different geographic landscapes. Thus, international trade is likely to occur in multiple aspects. Subsequently, [30,31] also argue that geographical indications, e.g., market access and protected designation of origin, have a remarkable impact on international trade.

However, market access influences international trade through economic gross domestic product (GDP) by firms regarding production. Another study, e.g., [32–34], emphasizes natural geography (climate change) for mitigation and adaptation through man-made geography, suggesting that geographic factors such as climate change and urbanization development have a joint effect on international trade. His research estimates potential trade-offs and conflicts between geographic factors (e.g., urbanization). Thus, trade-offs and conflicts have been gaining traction. The trade-offs are associated with natural and man-made geographic factors such as land, energy, transport, and green infrastructure. The findings conclude that mitigation measures could be impacted by increasing exposure to natural geographic factors (e.g., flooding and islands). Adaptation measures also enhance natural geographic factors (e.g., greenhouse gas emissions). However, our study contradicts this by exploring the simultaneous impact of natural and man-made geography on bilateral exports.

In addition, [35] argued that transportation costs have remarkable impacts on the bilateral exports. The reason is that transportation costs have direct correlation with the length of routes. Precisely, the transport costs are critical and heterogenous which are measured based on geographic locations such as mountainous area, desert and seashore, landscape as well. Moreover, [36] concluded that international trade and transport infrastructure are declining as a direct result of climate change. Climate change discourages trade flow in several countries. National climate change can indicate confusing decisions regarding the climate change's effect in international trade flows. The largest domestic market and more diversified trade pattern can absorb the climate shocks. Furthermore, climate change may reduce the comparative advantage thereby export level could be deteriorated. Subsequently, [37] showed that direct effect of climate change adversely affects the global supply chain and trade patterns. Moreover, extreme weather damages the transport infrastructure such as road, railway and maritime routes. Consequently, transportation costs could be higher for international trade.

Although studies related to climate change and international trade are few, qualitative agreement on climate change indicates that transport infrastructure would be balanced. The [38] demonstrated that climate change could imitate modes of transportation. Another study, e.g., [39] estimated the geographic structure of Ricardian into the general equilibrium for bilateral trade that are relevant to absolute and comparative advantages and geographical barriers for trade. Their implementations explore the probabilistic design of the heterogeneity of technology in which the model spreads naturally, several countries separated by geographical barriers. This design hints at a flexible framework for operating the geographical structure in the general equilibrium analysis. By extending the model for trade, which has significant application to the sensitivity of trade to input costs and geographic barriers. Moreover, given the goods, the location and its impact on input costs can play an important role. Potential gains from trade are created by comparative advantage.

Author [40] concluded that economic activities geographically shifted in Japan because of the migration of skilled workers or entrepreneurs who are associated with the silk industry

after trade openness and that the different structure of the present economic geography of Japan is prejudiced toward the east. Economic activities that have shifted to the eastward origin within Japan could have started at the beginning of the port era during the middle of the 19th century, which is called the "missing quarter-century." The significance of textiles and the silk industry at domestic and international trade levels is encouraged by the specific characteristics of Japan's economy and geography around the turn of the 19th century.

The findings demonstrate that Japanese silk fabric production and skilled labor may improve when dispersed between two regions, similar to the East and the West in an autarky that is consistent with historical observations. At Autarky, firms work in western regions, although they have a disadvantage in location in terms of raw material prices because this region has large local markets. Consequently, firms located in the West exit the market and enter markets in the East that have migrant skilled workers. In the East, a new spatial equilibrium takes place with international trade. Economic activities and mobile resources are more agglomerated in the east because of Japan's opening up during the missing quarter-century, encouraging the economic center of production in east Japan.

## Nexus between CO2 emissions and trade

This strand of literature describes the correlation between carbon dioxide ($CO_2$) emissions and trade. A substantial number of studies analyzes the effect of trade on carbon dioxide ($CO_2$) emissions for bilateral, multilateral and region to region. For instance, [41–43] argue that trade openness upsurges the $CO_2$ emission level. Precisely, exports contribute to growth in greenhouse emissions, however, signing the trade agreement between countries decline the influence of trade on $CO_2$ emissions. Furthermore, imports also have positive impacts on carbon emissions, although exports cause the reduction in carbon emissions after signing the trade agreements.

Subsequently, other studies also debate that trade openness cause augment the greenhouse effect in the short and long runs. Another study by [44,45] investigated the role of trade in environment, suggesting that global trade has worsened environmental pollution effects. The reason behind that trade openness upsurges the export level which causes the increase domestic production via expansion in scale of industries. Consequently, mass energy consumption is required to achieve the level of output and upsurged $CO_2$ emissions through economic activities. Subsequently, [46] analyze the $CO_2$ emissions embodies in bilateral trade between Japan and China, demonstrating that trade volume has a remarkable impact on $CO_2$ emissions in positive inclination.

In the contrast, [47] argue that trade openness can have negative effects on $CO_2$ emissions in OECD and industrialized countries respectively. Moreover, [48] discussed that trade openness may drastically adversely affect pollutants, especially energy. Besides, [49,50] also investigate that trade has a reticence effect on carbon dioxide emissions in OECD countries. However, it has positive impact on emissions in non-OECD countries. Conversely, [51] argue that trade openness drastically reduces per capita $CO_2$ emissions in Korean economy, precisely, this negative relationship scale effect, technique effect and structural effect. However, mitigation effect is governed by technical effect for nexus between trade and $CO_2$ emissions. It is worth notable that studies did not find the consistent accord on the correlation concerning the trade and $CO_2$ emissions.

Aforementioned literature emphasizes on the association between $CO_2$ emissions and trade openness from international perspective. Meanwhile, a scanty studies has indicated that aggregate global trade cannot determine the separate effects of exports and imports on $CO_2$ emissions [52–56]. Therefore, Hasanov et al. (2021) investigate the impact of exports and

imports on CO2 emissions separately, suggesting that imports have positive correlation with consumption-based CO2 emission, whereas exports can mitigate the domestic CO2 emissions.

## Nexus between trade and trade agreements

This aspect of the literature sheds light on the effects of trade agreements on trade from international perspective. Therefore, previous studies, e.g., [57–59] find that trade agreements have substantial impact on bilateral trade among the trading partners. Their findings suggest that particularly agricultural products are firmly restricted under the WTO's rules and regulation in order to protect the local industries. Authors [60] argues that preferential trade relationships are dissimilar with membership in the WTO due to legitimate groundworks. Precisely, with the support of free trade agreements (FTAs), multilateral trading system is much concerned for compatibility and radical changes under the framework of Article 24 GATT. Thus, trade agreements enrich the trade volume through the framework of WTO articles.

Subsequently, [61] debate on economic provision of preferential trade agreements to analyze the intensive and extensive margin bilateral exports. They suggest that PTAs have remarkable impact on exports with respect to time. Furthermore, measures applied at the border and behind the border enlarge different trade relationships. In addition, [62] also argues that regional trade agreements (bilateral, ASEAN, SAFTA) improve export efficiency over time. Furthermore, free trade agreements (FTAs i.e., SAFTA and ASEAN) bilateral agreements are most influencers and significant on exports with compared to PTAs (APTA & MERCOSER).

Another study, e.g., [63,64] analyze the impact of trade agreements with international oligopoly using three-country model. Their findings suggest that trade agreements affect the country's welfare through international trade system. However, non-member country is excluded from the effect of agreements. The author[65] also investigates the influence of provincial trade pacts on trading commodities. Findings suggest that regular trading partner's preferential import share is decreased by 29.8%, which is outsized and substantial for net-importing countries. Conversely, [66] analyze the effects of geographic-oriented trade agreements on trade. Their findings document that trade agreements could not clue marginal exports above and beyond the general exporting effect of trade agreements due to legal protection of geographical indications.

The above literature demonstrates that previous studies did not focus on the simultaneous impact of geographic factors, such as natural and man-made factors, on bilateral exports, especially in Asian countries. The simultaneous impact of geographic factors could not be discussed and linked with the GDP in previous studies. Precisely, an increase in GDP may affect the bilateral exports through geographic factors such as distance, access to Ocean and so on. In addition, researchers have extensively analyzed the nexus between trade openness and CO2 emissions for the largest trading and CO2 emission producing countries; however, some undeniable breaches exist in the literature.

Therefore, as best our knowledge, a very few studies focus on the effect of climate change (CO2 emissions) on bilateral exports in Asian countries. In the contrast, it is essential to analyze the effect of CO2 emissions on bilateral exports. The reason may behind that CO2 is an indicator of climate change that strongly influence the geographic-oriented environment. Consequently, exports may affect due to production costs, shortage of natural resources etc. Another gap motivates to analyze the effect of trade agreements on bilateral exports. Therefore, a scanty study considered the trade agreements such as WTO[x] and PTAs based on geographic factors.

Thus, this study investigates the impact of trade agreements, the reason may behind that trade agreements are diverse in nature, and sign based different reasons such as reduction in tariff, quantity restriction, labor mobility, capital mobility as well as political diplomacy.

However, this study argues that trade agreements may be sign based on geographic factors such as distance, access to Ocean, common border and so on.

## Methodology

### Theoretical framework and model

After reviewing the aforementioned literature on the effect of geographic factors on international trade, we developed a theoretical framework to estimate the influence of geographic factors on bilateral exports. A number of research studies argue that countries are giving great attention to export strategies to enhance trade volume and other economic goals. Specifically, exports are influenced by several factors, such as population, agglomeration, trade blocks, distance, and so on. However, geographic factors are the most important for bilateral exports. The reason may be that trading partners are located in different locations with diverse geographic factors.

More analytically, the simultaneous effect of geographic factors may also influence bilateral exports, indicating that the joint effect of climate change ($CO_2$ emissions), the ocean, and common border has a remarkable effect on bilateral export levels between the trading partners. To do so, bilateral exports as dependent variables, and geographic factors, trade agreements, and economic GDP as explanatory variables are used in the current analysis. Therefore, trade agreements also affect bilateral exports by eliminating the distortions that create the trade effect. In contrast, the imposition of distortions on member and non-member nations of any trade block creates trade diversion. Consequently, trade can be diverted from any equilibrium position.

The explanations for the variables are presented. Bilateral exports from country i to country j are measured in million dollars per year. The explanatory variables are Carbon dioxide ($CO_2$) emissions are the released emissions in tons from all sectors of the economy, such as agriculture, industry, or manufacturing sectors, per year. $CO_2$ emissions have a remarkable effect on international trade. The reason may be that these variable influences climate change and changes the atmosphere, resulting in the firm's decisions regarding the geographic-oriented environment or atmosphere. Consequently, the bilateral export level is enhanced between the trading partners [67].

Physical distance is defined as the distance from one place (capital to capital) in terms of a kilometer of a country, i.e., Ocean access is defined as a country that naturally has access to an ocean or sea. The common border is shared by two countries by land. These are key variables that are on the right side of equation (5). Moreover, we use proxies (e.g., $CO_2$ emissions and physical distance) to understand the concept of natural and man-made geography, which are estimated for first-nature and second-nature geography because these are the main factors of geography.

In addition, trade agreements defined as a dummy variable are included in the current model, if two member countries sign a preferential trade agreement is equal to one, otherwise zero [68] also argues that trade agreements (i.e., preferential trade agreements) are used for trade creation effects among the member countries. A number of studies investigated the potential trade diversion effects of preferential trade agreements by using the binary variables [69]. Therefore, member countries reduce trade restrictions such as tariffs and quotas to enhance trade volume. Thus, it is expected to have a positive effect on bilateral exports.

### Model estimation

$$ln\left(EXP_{ijt}\right) = \beta_0 + \beta_1 ln\left(CO2_{it}\right) + \beta_2 ln\left(PhyDIST_{ij}\right) + \beta_3 ln\left(AOij\right) + \beta_4 ln\left(CB_{ij}\right)$$
$$+ \beta_5 ln\left(GDP_{it}\right) + \delta_1 ln\left(Pop_{it}\right) + \delta_2 ln\left(distance\right) + \delta_3 tariff_{it} + \varepsilon_{it} \tag{1}$$

Equ. ([1]) explains that beta (β) and delta (δ) are explanatory and control variables respectively, $EXP_{ijt}$ indicates the exports between the origin country to the destination country, $CO2_{it}$ is the carbon dioxide emissions of a country $i$ in year $t$, $PhyDIST_{it}$ is the physical distance of a country $i$ in year $t$, $AO_{it}$ is access to Indian or Pacific Ocean of country $i$ in year, $CB_{it}$ is the common border of country $i$ with country $j$ in year $t$, $GDP_{ijt}$ demonstrates the GDP of country $i$ and $j$ in year $t$, and $Pop_{ijt}$ is the population of the exporting countries. Distance is measured in kilometers from the capital to the capital of the trading country. Mills ratio is the coefficients of the Heckman model (1979).

In addition, we extend the empirical model by adding the trade agreement and carbon dioxide emissions in the Equ. ([2]) to analyze the effect of simultaneous impact of trade agreement and carbon dioxide emissions (CO2) on bilateral exports. Thus, the empirical model is following:

$$ln\left(EXP_{ijt}\right) = \beta_0 + \beta_1 ln\left(CO2_{it}\right) + \beta_2 ln\left(PhyDIST_{ij}\right) + \beta_3 ln\left(AOij\right) + \beta_4 ln\left(CB_{ij}\right)$$
$$+ \beta_5 ln\left(GDP_{it}\right) + \beta_6\left(PTA_{ijt}\right) + \beta_7\left(GDP_{it}^2\right) + \beta_8\left(GDP \star CO2_{it}\right) + \beta_9\left(WTOX_{ijt}\right) + \varepsilon_{it} \tag{2}$$

Equ. ([6]) shows that export (s) is a function of carbon emissions (CO2), trade agreement (TDA), access to Ocean (AO), common border (CB), gross domestic product (GDP), non-square of economic GDP, and interaction term of economic GDP and CO2 (GDP*CO2). More specifically, $PTA_{ijt}$ denotes preferential trade agreement between country 'i' and country 'j' over time, whereas, the GDP*CO2 represents the joint effect of economic GDP and carbon emissions on bilateral exports. In addition, the $GDP_{it}^2$ indicates the non-linear effect of economic GDP on bilateral exports. $WTOX_{ijt}$ indicates the four-dimensional (dummy) variable with 0 or 1 for each WTOX provision.

## Data

The panel dataset covers the 46 Asian countries that collect information period from 2000 to 2020 on suitable variables through several international organizations (e.g., the World Bank, CEPII, and the United Nations Conference on Trade and Development (UNCTAD). The bilateral export data is collected by UNCTAD. The data on natural geography variables from World Development Indicators (WDI) and the man-made geography variables from CEPII are collected. The information on a country's characteristics (common border, landlocked, distance, GDP, and population) is collected by the WDI. In addition, the trade agreement (PTA)'s data are collected from WTO website, and complemented with information from Baier et al. (2008)3. PTA dummy variable refers to one if exporter and importer listed were in any one of PTA's listed either the WTO website. Besides, the data on CO2 emissions are collected from the World Input-Output Database (WIOD4) website. Moreover, the further detail is explained in the [Table 1].

## Estimation method

Our concern is to estimate the role of geographic factors, climate change and trade agreements in determining export activities at the bilateral level. The literature has demonstrated that natural geographic factors have a direct and indirect impact on international trade and infrastructure, as well as production and consumption [10,70,71]. Similarly, in man-made geography, transportation costs create economic activities in terms of adopting different strategies, such as substituting export strategies and increasing returns to scale, especially in the manufacturing sector.

We employ the structural gravity model [72–74] to investigate the influence of geographic factors, climate change and trade agreements on bilateral exports for Asian countries. In the

**Table 1. Variables and data source.**

| Variable | Measure | Code | Source |
|---|---|---|---|
| Exports | Ratio of export value to export price index | EXP | UNC-TAD |
| Distance | Kilometre from capital to capital | Dist | CEPII |
| Tariff | Weighted mean for all product in term of percentage | TRF | WDI |
| Access to Ocean | access to Indian or Pacific Ocean that defined as the equal to 1 if a country has access to Indian or the Pacific Ocean otherwise zero. | AC | CEPII |
| Common border | Dummy variable, 1, if a country shared border with its trading partner, otherwise zero | CB | CEPII |
| Carbon dioxide emissions | Measured in tons released from all sectors | CO2 | WIOD |
| Population | Total population measured in million | Pop | WDI |
| Gross domestic product | Gross domestic product measured in billion US dollar | GDP | WDI |
| Preferential trade agreement | if exporter and importer listed were in any one of PTA's listed either the WTO website | PTA | WTO |
| WTOX | four-dimensional (dummy) variable with 0 or 1 for each WTOX provision. | WTOX | WTO |

Source: author's calculations.

contrast, several studies, e.g., [75–81] used standard gravity equations to analyze the influence of trade barriers in trade flows.

Thus, this instrument is considered the most famous in foreign trade flows, such as imports and exports from origin to destination and vice versa, physical distance, economic mass (GDP), GNP, population size, culture, common language, and common border. To do so, econometric techniques have been employed, such as the Poisson Pseudo-Maximum Likelihood (PPML) to incorporate the issue of zero trade value in the dataset. The reason behind that some trading partners do not trade between each other such as Pakistan and Israel due to lack of diplomatic conflict or any economic and statically issue, which miss the observations [82–84]

Furthermore, this approach is also used to tackle the issue of heterogeneity, which could be exist. Subsequently, fixed effect technique is used to estimate the constant or fixed relationship in the explanatory variables. Random effect analyses the data to evaluate certain factors which casually affect the outcomes across the individual or groups. Afterward, Iterative reweighted least squares and quantile regression are used to correct the robustness check. The reason behind that extreme values can create biasness in the analysis process.

## Empirical results

**Descriptive statistics.** The descriptive statistics is analzyed in the Table 2 for measuring the central tendency, rationality and effective technique. Analysis shows that the observations are almost closed to their means for each variable, suggesting that the variation is found moderate. For instance, the logarithm means and S.D values of the variable export are 6.045 and 6.774 respectively, subsequently, the logarithm means and S.D values of distance are 3.612 and 3.366, which indicate that lowest variation is existed. However, the CO2, Pop, and GDP variables have difference between mean and S.D values, indicating that there is some variation in these observations. Besides, there are much differences between maximum and minimum values for the variable's exports, CO2, distance, population and GDP over time.

**Empirical analysis.** We use different econometric techniques to incorporate the influence of geographic factors on bilateral exports. Table 3 reveals the estimates of PPML, OLS, FE, and RE on domestic and partner geographic factors in bilateral exports. The findings indicate that geographic factors have a positive impact on bilateral exports. The domestic log of CO2

**Table 2. Central Tendency.**

| Variable | Obs | Mean | S.D | Min | Max |
|---|---|---|---|---|---|
| Exports | 44436 | 6.045 | 6.774 | 3.000 | 8.210 |
| Distance | 44436 | 3.612 | 3.366 | 1.856 | 4.040 |
| Tariff | 44436 | 0.892 | 0.753 | -1.523 | 1.574 |
| Access to Ocean | 44436 | 0.215 | 0.312 | 0.000 | 1.000 |
| Common border | 44436 | 0.248 | 0.305 | 0.000 | 1.000 |
| $CO_2$ | 44436 | 5.505 | 6.029 | 2.208 | 7.013 |
| Population | 44436 | 7.960 | 8.412 | 5.458 | 9.143 |
| GDP | 44436 | 11.601 | 12.076 | 6.909 | 13.049 |
| PTA | 44436 | 0.210 | 0.012 | 0.000 | 1.000 |
| WTOX | 44436 | 0.010 | 0.032 | 0.000 | 1.000 |

Note: all variables are transferred into logarithm.

declined below the export level and significantly increased under FE and RE. Comparatively, the importing' log of CO2 is a positive inclination factor for bilateral export levels except under OLS.

$CO_2$ has a high magnitude for importing countries than exporting countries. Both countries substantially contribute to determining bilateral exports. Subsequently, physical distance is found to be a statistically significant factor with high magnitudes. Geographic factors from exporting countries have a greater impact than importing countries. The findings show that geographic factors for exporting countries influenced bilateral exports by 72.2% more than importing countries. By controlling the country-specific effect, bilateral exports are increased by 22.6% because of the margin of countries. Therefore, domestic physical distance has a stronger effect than physical distance and $CO_2$. The author [85,86] found that the firm's decisions about movement are made based on the specific location where high demand market size exists after decreasing transportation costs.

Supplementary natural geographic factors, such as access to the ocean and common borders, significantly influence the export level. Bilateral exports are enhanced because of a high contribution of access to the ocean from the importing country's countryside, resulting in high magnitudes. Countries that have access to the ocean benefit during trading of products through air infrastructure. However, exporter geographical countries are also geographically benefitting because they have a common border. Having a common border reduces trade costs through bilateral infrastructure [67]. The outcomes from PPML indicate that explanatory variables have remarkable impact on exports. The estimation is consistent rather than traditional OLS, fixed and random effect. The reason behind that this technique incorporates the zero trade issues in bilateral exports. More precisely, some trading partners do not report or trade between each other such as Pakistan and Israel due to diplomatic conflicts. Thus, zero trade issues is occurred. Empirical evidence shows that a 1% increase in CO2 upsurges exports by 10.2%. The reason may behind that extreme weather stimulates production of dissimilar products, which are transported from/to multiple locations based on environment. The coefficient of distance is negatively correlated with exports, 76.2% decrease in exports is due to a one-unit change in distance. Likewise, rest of the trade barriers are also found negative correlated with exports and statistically significant at 1% and 5% levels.

Our main concern is the simultaneous impact of geographic factors on bilateral exports, reported in Table 4. We perform a two-stage sample selection [37]. Columns (1)–(4) indicate domestic geographic estimates, whereas Columns (5)–(8) indicate importing geographic estimates.

**Table 3. Regression analysis.**

| ___-Estimation | Exporting countries | | | | Importing countries | | | |
| --- | --- | --- | --- | --- | --- | --- | --- | --- |
| | PPML | OLS | Fixed Effect | Random Effect | PPML | OLS | Fixed Effect | Random Effect |
| | (1) | (2) | (3) | (4) | (1) | (2) | (3) | (4) |
| VARIABLES | *Dependent variable: exports* | | | | | | | |
| $CO_2$ | 0.102* | 0.020 | -0.287 | 0.132 | 0.091*** | 0.068 | 0.085** | 0.084** |
| | (0.021) | (0.054) | (1.119) | (0.477) | (0.021) | (0.058) | (0.040) | (0.040) |
| Physical-distance | 0.671** | 0.859*** | 0.414*** | 0.525*** | 0.541** | 0.127*** | 0.188*** | 0.187*** |
| | (0.012) | (0.016) | (0.082) | (0.072) | (0.021) | (0.018) | (0.012) | (0.012) |
| Lndistance | -0.762*** | -0.840*** | -0.857*** | -0.858*** | -0.671** | -0.994*** | -1.241*** | -1.239*** |
| | (0.001) | (0.030) | (0.021) | (0.029) | (0.020) | (0.031) | (0.023) | (0.024) |
| Access to Ocean | 0.201** | 0.751*** | – | 0.622 | 0.210*** | 3.006*** | – | 3.048*** |
| | (0.021) | (0.044) | – | (0.386) | (0.027) | (0.039) | – | (0.404) |
| Common border | 0.021*** | 2.416*** | 2.609*** | 2.606*** | 0.032*** | 2.248*** | 1.501*** | 1.506*** |
| | (0.091) | (0.075) | (0.082) | (0.081) | (0.032) | (0.080) | (0.066) | (0.066) |
| Lngdp | 0.329** | 0.281*** | 0.429*** | 0.445*** | 0.450*** | 0.644*** | 0.582*** | 0.582*** |
| | (0.017) | (0.016) | (0.041) | (0.037) | (0.030) | (0.018) | (0.013) | (0.013) |
| lntariff | -0.039** | -0.366*** | -0.257*** | -0.266*** | -0.023*** | -0.358*** | -0.379*** | -0.379*** |
| | (0.061) | (0.022) | (0.052) | (0.051) | (0.032) | (0.023) | (0.018) | (0.017) |
| lnpop | 0.035*** | 0.046*** | 0.771*** | 0.324*** | 0.045*** | 0.267*** | 0.303*** | 0.303*** |
| | (0.013) | (0.016) | (0.174) | (0.103) | (0.054) | (0.015) | (0.013) | (0.011) |
| Constant | 1.203** | -1.818** | -8.564 | -8.226 | 1.4321** | -5.726*** | -3.407*** | -5.266*** |
| | (0.021) | (0.718) | (13.700) | (6.104) | (0.042) | (0.757) | (0.542) | (0.628) |
| Observations | 44436 | 44436 | 44436 | 44436 | 44344 | 44436 | 44436 | 44436 |
| R-squared | | 0.405 | 0.104 | – | | 0.349 | 0.426 | – |
| Number of exporters | 46 | – | 46 | 46 | 46 | – | 46 | 46 |
| Country effect | Yes | – | – | – | Yes | – | – | – |
| Time effect | Yes | – | – | – | Yes | – | – | – |

**Note:** the term ***, ** and * denote that significance level at 1%, 5% and 10% respectively. Ln indicates logarithm for variables.

The findings exhibit that exporting countries' CO2 has a higher impact on bilateral export levels and probability than the importing country's geography. Similarly, exporting countries' physical distance has significantly impacted bilateral exports under outcome and selection equations.

The export level and its probability are declined, resulting in the negative simultaneous impact of geography and the GDP. The marginal export level of exporting county's $CO_2$ has a positive impact by 247.6% and is greater than importing countries' geography, conditionally varying the GDP level. Similarly, the marginal export probability of exporting country's $CO_2$ also has a positive impact of approximately 191.6% more than that of importing countries. Furthermore, the marginal export level and the marginal probability of physical distance are highly affected by approximately 264.09%, 204.56%, 14.494%, and 93.04% for both exporting and importing countries' geography due to GDP.

On conversely, [87] argue that trade is a benign effective factor for the environment in the short-run but harmful in the long-run. Moreover, $CO_2$ emissions are negatively affected by urbanization in the short and long run. However, we find that the simultaneous impact of geographic factors has negative coefficients under outcome and selection equations. The complementary of CO2 and physical distance strongly impact the decline of the export level and probability between trading countries.

**Table 4. Simultaneous Effects of Geographical Factors on Asian's Bilateral Exports.**

| Location | Exporting Countries | | | | Importing Countries | | | |
|---|---|---|---|---|---|---|---|---|
| Estimation | GDP Interaction | | Geographical Interaction | | GDP Interaction | | Geographical Interaction | |
| | (1) | (2) | (3) | (4) | (5) | (6) | (7) | (8) |
| *Dependent Variable: Export$_{ijt}$* | | | | | | | | |
| $CO_2$ | 5.281** | 4.033*** | 4.372*** | 2.755*** | -0.348 | 0.858 | 1.554** | 0.494* |
| | (2.437) | (0.695) | (1.276) | (0.316) | (1.132) | (0.626) | (0.638) | (0.276) |
| Physical-distance | 0.969* | 0.794*** | 4.498*** | 3.024*** | 0.102 | 0.539*** | 1.333** | 0.519* |
| | (0.512) | (0.0879) | (1.342) | (0.332) | (0.230) | (0.0744) | (0.650) | (0.288) |
| Lndistance | -0.254*** | -0.120*** | -0.265*** | -0.125*** | -0.394*** | -0.193*** | -0.472*** | -0.201*** |
| | (0.062) | (0.016) | (0.051) | (0.016) | (0.057) | (0.016) | (0.053) | (0.016) |
| Access to Ocean | 1.168*** | 0.641*** | 1.218*** | 0.658*** | 1.769*** | 1.447*** | 2.346*** | 1.460*** |
| | (0.310) | (0.033) | (0.231) | (0.033) | (0.421) | (0.032) | (0.364) | (0.032) |
| Common Border | 0.425 | 0.634*** | 0.437** | 0.619*** | 0.175 | 0.607*** | 0.397*** | 0.573*** |
| | (0.275) | (0.040) | (0.195) | (0.039) | (0.168) | (0.039) | (0.141) | (0.039) |
| LnGDP | 2.659** | 2.062*** | 0.215*** | 0.089*** | 0.147 | 0.951*** | 0.586*** | 0.340*** |
| | (1.246) | (0.327) | (0.043) | (0.014) | (0.608) | (0.294) | (0.087) | (0.013) |
| Lntariff | -0.050*** | -0.032*** | -0.052*** | -0.033*** | -0.025*** | -0.018*** | -0.031*** | -0.017*** |
| | (0.016) | (0.002) | (0.011) | (0.002) | (0.005) | (0.002) | (0.005) | (0.002) |
| Lnpopulation | -0.202*** | -0.086*** | -0.223*** | -0.095*** | 0.044** | 0.056*** | 0.056*** | 0.039*** |
| | (0.044) | (0.010) | (0.038) | (0.010) | (0.021) | (0.009) | (0.017) | (0.009) |
| $CO_2$*GDP | -0.183** | -0.146*** | | | 0.025 | -0.030 | | |
| | (0.089) | (0.026) | | | (0.043) | (0.024) | | |
| PhyDist*GDP | -0.018 | -0.016*** | | | -0.002 | -0.020*** | | |
| | (0.013) | (0.003) | | | (0.008) | (0.002) | | |
| $CO_2$*PhyDist | | | -0.325*** | -0.217*** | | | -0.105** | -0.041* |
| | | | (0.102) | (0.026) | | | (0.053) | (0.023) |
| Lambda | 1.314** | | 1.377*** | | 0.755** | | 1.292*** | |
| | (0.603) | | (0.430) | | (0.380) | | (0.323) | |
| Constant | -63.360* | -59.750*** | -46.940*** | -39.170*** | 6.944 | -26.900*** | -20.560** | -16.190*** |
| | (35.420) | (8.544) | (17.380) | (3.905) | (16.390) | (7.590) | (9.317) | (3.410) |
| Observations | 44436 | 44436 | 44436 | 44436 | 44436 | 44436 | 44436 | 44436 |

**Note:** the term ***, ** and * denote that significance level at 1%, 5% and 10% respectively. Ln indicates logarithm for variables.

The authors [39] also documented their findings, the geographic structure of Ricardian into the general equilibrium, and the proposed simple structural equations for bilateral trade, which are relevant to absolute and comparative advantages and geographical barriers for trade. However, they did not cover the geographic aspect as we did. The marginal export level and the marginal probability of emissions are found significant and positive concerning physical distance. Similarly, the marginal export level and the marginal export probability of physical distance have a positive impact, which is less than domestic geography. Variables such as access to an ocean and a common border have a significant and strong impact on bilateral exports and their probability among the Asian countries. For instance, Central Asia, West Asia, and South Asia, excluding Sri Lanka, the Maldives, China, and Russia, have more common borders and access to an ocean. The study concluded that all geographic factors and the GDP are complementary to one another in determining the export level and its probability between trading partner countries under both dimensions.

The estimates of the structure gravity model and multilateral resistance terms for geographical factors are reported in Table 5. We perform the structure gravity model and the two-stage sample selection [37] to examine the influence of trade barriers and the correlation of unobserved variables among the explanatory and error terms, respectively. The gravity variable, distance, is found with a negative and significant magnitude that reduces the export level in both dimensions (domestic and partners), but the partner countries' distance is greatly affected compared with that of the domestic country.

The main geographic factors have a significant influence on the bilateral export level and its probability. During bilateral exports, access to an ocean is found to have a greater impact on exports in the importing country's geography than the exporting country's geography. Access to an ocean and common border geographical factors also positively influence the bilateral exports between trading partner countries. The country characteristic variables, such as population, decline in the export level and probability in exporter countries but increase in

**Table 5. A Structural Gravity with Trade Agreement and CO2 Emissions.**

| Variable | Exporting Countries Model | | | Importing Counties Model | | |
|---|---|---|---|---|---|---|
| | Linear | Non-linear | Interaction | Linear | Non-linear | Interaction |
| $CO_2$ | -0.250 | 0.176** | 0.117*** | 0.132 | 0.152*** | 0.169** |
| | (0.045) | (0.051) | (0.054) | (0.110) | (0.041) | (0.021) |
| Physical-distance | -0.125*** | -0.130*** | -0.228*** | -0.186*** | -0.179*** | 0.140*** |
| | (0.001) | (0.020) | (0.009) | (0.003) | (0.001) | (0.007) |
| Access to Ocean | 0.151*** | 0.321** | 0.622* | 1.006*** | 0.153*** | 0.108*** |
| | (0.001) | (0.021) | (0.086) | (0.002) | (0.002) | (0.030) |
| Common border | 0.627*** | 0.017* | 0.142*** | 0.197*** | 0.104*** | 0.054*** |
| | (0.011) | (0.014) | (0.030) | (0.080) | (0.032) | (0.031) |
| LnGDP | 0.107*** | 0.132** | 0.121*** | 0.372*** | 0.032* | 0.043*** |
| | (0.001) | (0.032) | (0.032) | (0.102) | (0.021) | (0.032) |
| PTA | 0.114*** | 0.204*** | 0.109*** | 0.140*** | 0.128*** | 0.157*** |
| | (0.001) | (0.023) | (0.011) | (0.015) | (0.036) | (0.011) |
| CPM | 0.242** | – | – | 0.312*** | – | – |
| | (0.002) | – | – | (0.001) | – | – |
| CPP | 0.111*** | – | – | 0.091*** | – | – |
| | (0.020) | – | – | (0.031) | – | |
| LMB | – | 0.190** | – | – | 0.173*** | – |
| | – | (0.004) | – | – | (0.023) | – |
| EVS | – | – | 0.201*** | – | – | 0.131*** |
| | | – | (0.010) | – | – | (0.004) |
| $GDP^2$ | – | 0.102*** | – | – | 0.104*** | – |
| | – | (0.030) | – | – | (0.020) | – |
| GDP*CO2 | – | – | -0.035*** | – | – | -0.013*** |
| | – | – | (1.070) | – | – | (0.051) |
| Constant | 1.971*** | 0.564** | 0.126** | 0.486* | 0.214 | 0.127*** |
| | (0.047) | (0.070) | (0.100) | (0.028) | (0.041) | (0.308) |
| R-squared | 0.145 | 0.104 | – | 0.449 | – | – |
| Number of exporters | – | 46 | 46 | – | 46 | 46 |

**Note:** the term ***, ** and * denote that significance level at 1%, 5% and 10% respectively. Ln indicates logarithm for variables.

importer countries. The remoteness in exporter countries seems with higher magnitudes that cause a reduction in the export level and probability than importer countries.

More interestingly, the effect of a trade agreement is also investigated on bilateral exports along with the non-linear and simultaneous effect of economic GDP. The outcomes are reported in Table 6. A trade agreement has a remarkable impact on bilateral exports between the trading partners. More specifically, trade agreements provide trading partners with benefits in the form of nominal (i.e., tariff) or quantity (i.e., imposing quantity restrictions). The outcomes reveal that a 1% increase in PTA upsurges bilateral exports by 11.4% in the case of domestic geographic countries. On the contrary, 14% of bilateral exports are increased due to a 1% change in trade agreements for the partner geographic countries (importers). This suggests that trade agreements between the trading partners reduce the tariff or enhance export quantity. Consequently, bilateral exports have increased over time. However, trade agreements are diverse in nature, which can have a substantial impact on bilateral exports due to different trade mechanisms. Furthermore, trade agreements are considered based on a geographic-oriented environment where production and consumption have contrarily

**Table 6. Structural Gravity Model.**

|  | Exporting Countries | | | Importing Countries | | |
|---|---|---|---|---|---|---|
|  | (1) | (2) | (3) | (4) | (5) | (6) |
| VARIABLES | Dependent Variable: Exports$_{ijt}$ | | | | | |
| $CO_2$ | -0.355 | 0.376** | 0.327*** | 0.167 | 0.352*** | 0.069** |
|  | (0.245) | (0.151) | (0.054) | (0.119) | (0.046) | (0.030) |
| Physical-distance | 0.145 | 0.330*** | 0.428*** | 0.186 | 0.379*** | 0.249*** |
|  | (0.201) | (0.121) | (0.010) | (0.143) | (0.050) | (0.007) |
| Lndistance | -0.891*** | – | – | -1.239*** | – | – |
|  | (0.176) | – | – | (0.168) | – | – |
| Access to Ocean | 0.751*** | – | 0.622 | 3.006*** | 1.553*** | 1.308*** |
|  | (0.044) | – | (0.386) | (0.039) | (0.282) | (0.032) |
| Common border | 2.627*** | 0.117 | 0.742*** | 1.497*** | 0.704*** | 0.854*** |
|  | (0.401) | (0.201) | (0.035) | (0.283) | (0.166) | (0.034) |
| LnGDP | -0.107 | – | – | 0.572*** | – | – |
|  | (0.094) | – | – | (0.192) | – | – |
| Lntariff | -0.124 | -0.214*** | -0.189*** | -0.340*** | -0.328*** | -0.257*** |
|  | (0.076) | (0.056) | (0.011) | (0.071) | (0.053) | (0.011) |
| Lnpopulation | -0.097 | -0.122*** | -0.089*** | 0.314*** | 0.104*** | 0.122*** |
|  | (0.511) | (0.030) | (0.009) | (0.077) | (0.027) | (0.008) |
| Remoteness | – | -3.875 | -6.335*** | – | 0.002*** | 0.001*** |
|  | – | (3.875) | (1.970) | – | (0.030) | (1.120) |
| Lambda | – | 0.242 | – | – | 0.740*** | – |
|  | – | (0.381) | – | – | (0.275) | – |
|  | – | – | – | – | – | – |
| Constant | 24.970*** | -8.564 | -8.226 | -3.486 | 1.294 | -8.327*** |
|  | (8.947) | (13.700) | (6.104) | (2.728) | (1.941) | (0.398) |
| Observations | 44436 | 44436 | 44436 | 44436 | 44436 | 44436 |
| R-squared | 0.145 | 0.104 | – | 0.449 | – | – |
| Number of exporter | – | 46 | 46 | – | 46 | 46 |

**Note:** the term ***, ** and * denote that significance level at 1%, 5% and 10% respectively. Ln indicates logarithm for variables.

existed. Previous studies (e.g., Hussain et al., 2020; Friel et al., 2020) also confirm our findings, and argue that trade agreements have a remarkable impact on bilateral exports between the trading partners.

In addition, WTO$^x$ provisions are positively incline towards bilateral exports. To avoid collinearity, each provision is investigated separately by not examined into single equations. However, each estimation has been measured with a dummy variable, suggesting that whether a PTA endures or not. The results demonstrate that the coefficient of capital mobility (CPM) clause in PTA has positive and significant impact bilateral exports, which implies that a 1% increase in CPM upsurges bilateral exports by 24.2% and 32.1% for exporting and importing countries models respectively. Subsequently, the coefficient of competition policy (CPP) has also significant positive impact on bilateral exports. It implies that 11.1% bilateral exports are increased due to a 1 percentage change in CPP for exporting countries model. Conversely, a 1% change in CPP upsurges bilateral exports by 9.1% for importing countries model. Labor mobility (LMB) provisions stimulate bilateral exports approximately 19%, suggesting that provision of facilities for movement cause an increase in export volume. It is worth notable that migration stock variable added as control variable and isolates the certain outcomes of labor mobility. Conversely, bilateral exports are increased about 17.3% due to a one-percentage point in LMB. The magnitude of environmental standards (EVS) has also positive effects on bilateral exports, which implies that 20.1% and 13.1% bilateral exports are due to one unit change in EVS for exporting and importing countries models respectively. The outcomes suggest that environmental standards in PTAs persuades efficiency, innovation and business effectiveness, ensuing in exports.

Considering the simultaneous effect of GDP and CO2 emissions, the coefficient of interaction term (GDP*CO2) adversely affect the bilateral exports. This implies that a 1% change in GDP*CO2 reduces bilateral exports by 3.5% in the case of domestic geographic countries. On the contrary, the outcomes for the case of partner geographic countries also confirm the negative correlation with bilateral exports; a 1.3% decrease in bilateral exports is due to a 1% change in interaction terms. This suggests that economic GDP and CO2 emissions are simultaneously deteriorating bilateral exports. The reason may be that an increase in economic growth causes carbon dioxide emissions, resulting in extensive production and consumption levels that directly or indirectly affect bilateral exports based on geographic locations. Furthermore, the marginal impact of bilateral exports with respect to CO2 emissions is positive for bilateral exports. The outcomes indicate that a 1% increase in economic GDP increases the marginal impact by 14.1%, suggesting that CO2 emissions affect bilateral exports through economic GDP over time.

**Robustness checks.** Geographical factors could also be determined by trade, the presence of natural resources, and human economic activities. Thus, the potential reverse causality problem could be presented. However, the large investment in human and natural geography to create economic activities and discover natural resources can provide great returns to countries through production and export channels. The expectations of causality on the side, export level, and export probability can be directly influenced by geographic factors compared with other ways [70,88]. The indicators of geography do not greatly change their values concerning time, especially natural factors, reflecting that any unexpected change in geographical factors has not happened because of export flow. Thus, we report the potential endogeneity issue by exploiting the lack of geography as a new measurement of distance.

Our analysis deals with heteroscedasticity (constant elasticity models) and checks the consistency of the baseline estimate with OLS and the pseudo-maximum likelihood estimator, as suggested by [89–91]. They explain that the lowest bias is produced by the Poisson Pseudo-Maximum Likelihood (PPML) for several heteroscedasticity patterns. The results are

reported in Table 7 for the robustness check. The replacement of the core gravity variable, distance, with a new measurement scale indicates the same results with the minor changes in magnitude as estimated in a simple regression. The remaining gravity variables do not change. Column (6) reports that all geographic variables are positive, with the exception of physical distance.

Physical distance as a geographical factor has a high impact on exports. Export volume is highly affected because of the great magnitude of domestic and partner countries' populations and GDPs. Columns 2 and 5 report the estimate of PPML. All geographical factors expect positive signs and are statistically more significant than domestic CO2. Domestic countries that have access to the Indian or the Pacific Oceans remain significantly negative compared with partner countries. The variable tariff imposed by domestic countries on trade policy has expected signs. The world's GDP is found to be greater in magnitude in domestic geography than in partner countries' geography.

**Table 7. Robustness Checks.**

| | Exporting Countries | | | Importing Countries | | |
|---|---|---|---|---|---|---|
| Estimation | QREG | PPML | 2SLS | New Variable | PPML | 2SLS |
| | (1) | (2) | (3) | (4) | (5) | (6) |
| VARIABLES | *Dependent Variable: Exports* | | | | | |
| $CO_2$ | -0.325 | -0.767*** | 1.069 | -0.327*** | 0.259*** | 7.399 |
| | (1.494) | (0.073) | (3.447) | (0.048) | (0.170) | (21.410) |
| Physical-distance | 0.552*** | 0.209*** | 0.939 | 0.117*** | 0.008*** | -0.086 |
| | -0.325 | (3.720) | (0.878) | (0.018) | (8.970) | (3.986) |
| LnGDP | 0.444*** | 0.548*** | 0.466 | 0.681*** | 0.663*** | 4.628 |
| | (0.055) | (1.880) | (0.800) | (0.019) | (9.180) | (9.142) |
| Lntariff | -0.366*** | – | -0.032*** | -0.023*** | -0.018*** | – |
| | (0.022) | – | (3.040) | (0.003) | (1.440) | – |
| Lndistance | -1.023*** | -0.712*** | – | – | -0.634*** | – |
| | (0.038) | (7.240) | – | – | (6.820) | – |
| Access to Ocean | -0.004 | – | – | 2.845*** | – | – |
| | (0.609) | – | – | (0.040) | – | – |
| Common border | 2.381*** | -0.179*** | – | 3.456*** | 0.106*** | – |
| | (0.109) | (1.730) | – | (0.071) | (1.750) | – |
| Lnpopulation | 0.789*** | 0.583*** | – | 0.182*** | 0.0417*** | – |
| | (0.233) | (5.510) | – | (0.015) | (5.240) | – |
| Lntotal distance | – | – | – | -0.302*** | – | – |
| | – | – | – | (0.054) | – | – |
| Lnworld GDP | – | – | 707.1* | – | – | -3.065 |
| | – | – | (6.600) | – | – | (6.573) |
| Constant | – | – | -22,418* | -6.808*** | – | 6.873 |
| | – | – | (13.521) | (0.973) | – | (2.757) |
| Observations | 44436 | 44436 | 44436 | 44436 | 44436 | 44436 |
| R-squared | 0.405 | 0.104 | – | 0.328 | – | – |
| Number of exporters | – | 46 | – | – | 46 | – |

**Note:** the term ***, ** and * denote that significance level at 1%, 5% and 10% respectively. Ln indicates logarithm for variables.

## Conclusion

This study investigates the effects of geographic factors, climate change and trade agreements on bilateral exports for Asian countries. To do so, we use the panel dataset for Asian countries period from 2000 to 2020 by employing the augmented gravity model. Findings demonstrate that countries better utilize the natural and man-made economic activities for the bilateral exports among their trading partners. More specifically, GDP and geographic factors are complementary in determining the export level and probability in both locations. Besides, the marginal impact of natural and man-made activities concerning the GDP is positively affected by adding the interaction terms in the model.

The marginal export level and marginal export probability concerning the geographic factors are greater than zero because of the additional change in natural and human-made economic activities. Countries that have access to the Indian or the Pacific Ocean and common borders can increase the export level and its probability between bilateral trading partners through establishing the transport infrastructure. Conventional gravity variables, such as the GDP and distance or remoteness, including control variables, have positively and negatively impacted bilateral exports and their probability, respectively. Remoteness reduces the export level in terms of charging high trade costs or transportation movement prices but presents it differently in the countries. Thus, the countries must improve their natural and man-made economic activities through optimal allocation to enhance the export level and probability at above average.

In addition, this study probes that whether preferential trade agreements are favorable or unfavorable to bilateral exports. PTAs are extensively diverse and difficult to treated any more in the empirical research as a homogenous variable, the reason may behind that expansion in international trade development, inclusion of new innovation and geographic factors and so on. Therefore, the clauses of WTOx namely labor mobility, capital mobility, competition policy and environmental standards were investigated, suggesting that each clause has positive impact on bilateral exports. Besides, climate change ($CO2$ emissions) has also positive impact on bilateral exports through three effects, scale effect, technique effect and structural effect. It is worth notable that exports can be increased after signing the trade agreements with regards climate change between the trading partners.

## Policy implications

We recommend policy implications: countries should productively utilize 196% on average their natural resources to enhance bilateral exports through the infrastructure connectivity in Asia. Countries with the largest land area should focus on mountains, deserts, temperature, and climate changes in terms of land use for developing a trade-related environment. For instance, China, Russia, Indonesia, India, Pakistan, and Iran should adopt strategies related to geographic factors (e.g., reduction $CO_2$ by 10.2%), provide better facilities at cross borders, and develop multilateral projects. Asian countries who share a common border can benefit from bilateral exports but require better facilities and must utilize geographic endowments. Similarly, Asian countries with islands or do not share borders, such as Sri Lanka, the Maldives, Japan, and the Philippines, must focus on 168.9% facilities concerning access to the ocean (seashore or ports).

Findings further suggest that natural geographic factors should be improved by 16.325% (estimated by structural gravity technique), resulting in relatively low magnitudes in domestic and partner countries. In the location of geographic factors in domestic (exporter) countries, export levels are affected because of country characteristics. In contrast, man-made geographic factors seem to have a relatively high impact. This suggests that man-made economic

activities are well performed compared with natural economic activities to increase the export level between Asian trading partners.

Climate change ($CO_2$ emissions) has 528% (estimated by simultaneous approach) detrimental effects on the ecosystem and are deteriorating the global environmental quality gradually. Despite their environmental effects, $CO_2$ emissions also have a remarkable impact on international trade approximately 167.6% on average, particularly bilateral exports. Countries should sign trade agreements and formulate the policy regarding geographic-oriented $CO_2$ emissions, indicating that the effects of $CO_2$ emissions are diverse from country to country, region to region, and geographic landscape. Consequently, bilateral exports are affected (which means that firms make decisions for production and export strategies based on $CO_2$ emission affected areas), and trade agreements are also changed in their terms and conditions in nature. Countries should control 10.7% economic growth that causes increase in $CO_2$ emissions through scale effect. However, structural effect can reduce $CO_2$ emissions through advancement in technology, which can lead bilateral exports between the trading partners.

In addition, PTAs, plurilateral trade agreements, WTOX clauses should be signed by each trading countries to augment 14.23% export level. More precisely, all trade agreements should be also based on geographic and economic factor including environmental standards.

This study presents the limitations to examine the simultaneous impact on the bilateral exports of geographic factors. For instance, the total land-use area by mountain, desert, seashore, and urbanization is not discussed in our model. More precisely, the Asia covers greatest mountainous area of the Earth's total land approximately 30% (44579000 squared kilometers) e.g., Himalayas, Karakoram Hindu Kush and Pamirs, which substantially engrosses to yield numerous agricultural products, suggesting that mountainous induces exports could be enhanced through export-oriented strategies. Although, mountainous area can raise the transport costs. Likewise, it is noteworthy that total desert area of Asia has been concealed by 32% (6344000 square mile) of the globally, i.e., Arabian and Gobi deserts, which sheds light on transport infrastructure development concern over increment of bilateral exports. Subsequently, Asia has also largest coastline area about 62800 kilometers (e.g., Arctic, Pacific and Indian Oceans) that substantially influences commercial-oriented infrastructure, use for trade-oriented activities along with coastal lines.

The agglomeration of economic activities, market access, and production costs are also not included. More precisely, constellation of economic activities, e.g., specific commercial or business and people areas, manufacturing units, industries based on location choice remarkably effect the bilateral exports by reducing the transportation costs, enhance the access to the suppliers, technology spillover, quality of the product and so on. Furthermore, market access (product or service selling ability of the company) and production costs (expenses are bearded by the producers) can have substantial influence on bilateral exports. The reason may behind that these factors drastically improve the efficiency and productivity of the export-oriented firms and companies. Thus, we can expect future research to cover land area by mountains, deserts, and seashores, which can be estimated to determine international trade, and the preservation of natural resources, market access, and man-made skills as substantial factors for economic development through trade channels.

## Author contributions

**Conceptualization:** Zahid Hussain.

**Data curation:** Qianxu Liang.

**Formal analysis:** Qianxu Liang, Zahid Hussain, Chaonan Wang.

**Funding acquisition:** Qianxu Liang.

**Investigation:** Zahid Hussain.

**Methodology:** Lin Shao.

**Resources:** Lin Shao, Yufang Chao.

**Software:** Yufang Chao.

**Supervision:** Yufang Chao, Haiying Liu.

**Validation:** Haiying Liu.

**Writing – original draft:** Haiying Liu, Chaonan Wang.

**Writing – review & editing:** Chaonan Wang.

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
