## [Decision Letter · Decision Letter 0]

27 Dec 2024

PONE-D-24-32100Climate Change, Geography and Trade Agreements: A Perspective of Asian Bilateral TradePLOS ONE

Dear Dr. Hussain,

Thank you for submitting your manuscript to PLOS ONE. After careful consideration, we feel that it has merit but does not fully meet PLOS ONE’s publication criteria as it currently stands. Therefore, we invite you to submit a revised version of the manuscript that addresses the points raised during the review process.

We look forward to receiving your revised manuscript.

Kind regards,

Asif Khan, PhD Law

Academic Editor

PLOS ONE

Journal Requirements:

Reviewers' comments:

Reviewer's Responses to Questions

**Comments to the Author**

1. Is the manuscript technically sound, and do the data support the conclusions?

Reviewer #1: Partly

Reviewer #2: Yes

2. Has the statistical analysis been performed appropriately and rigorously? 

Reviewer #1: No

Reviewer #2: Yes

3. Have the authors made all data underlying the findings in their manuscript fully available?

Reviewer #1: Yes

Reviewer #2: Yes

4. Is the manuscript presented in an intelligible fashion and written in standard English?

Reviewer #1: Yes

Reviewer #2: Yes

5. Review Comments to the Author

Reviewer #1: we written but it can be improved with more recent knowledge and citations. The author must start from most recent knowledge and information so that there must not be any broken research link. most recent developments must be included

Reviewer #2: Recommendations:

Based on the findings and conclusions of the study, the following minor recommendations can be made:

1.Expanding the Scope of the Analysis:

The authors acknowledge the limitations of the study and suggest that future research should incorporate additional factors, such as the impact of agglomeration, market access, and production costs, on bilateral exports (Section 6). Expanding the scope of the analysis can provide a more comprehensive understanding of the drivers of bilateral trade in Asian countries.

2.Incorporating Detailed Geographic Factors:

While the study examines the impact of various geographic factors, such as access to the ocean and common borders, the authors could consider incorporating more detailed geographic factors, such as the total land-use area by mountains, deserts, and seashores, to further enhance the understanding of their influence on bilateral exports (Section 6).

Overall, this is a well-designed and executed study that contributes to the understanding of the role of geographic factors, climate change, and trade agreements in determining bilateral exports in Asian countries. The authors have thoroughly addressed the research objectives and have provided a comprehensive analysis supported by appropriate empirical methods and robust findings.

6. PLOS authors have the option to publish the peer review history of their article (what does this mean? ). If published, this will include your full peer review and any attached files.

**Do you want your identity to be public for this peer review?** For information about this choice, including consent withdrawal, please see our Privacy Policy .

Reviewer #1: **Yes: ** Muhammad Imran Khan

Reviewer #2: **Yes: ** Faiza Choudhary

---

## [Author Response · Author response to Decision Letter 1]

5 Feb 2025

Author Response to Reviewer Comments

Article title: PONE-D-24-32100

Title: Climate Change, Geography and Trade Agreements: A Perspective of Asian Bilateral Trade

Dear Editor

Revision

We are extremely grateful for the valuable suggestions and comments from the reviewers. We have carefully addressed each reviewer comment. The changes made in the revised manuscript according to comments of the reviewer 1 are highlighted in Yellow, comments of the reviewer 2 are highlighted in Green.

5. Review Comments to the Author

Reviewer #1: we written but it can be improved with more recent knowledge and citations. The author must start from most recent knowledge and information so that there must not be any broken research link. most recent developments must be included

Response:

Thank you for your comments. We have substantially improved by adding the new knowledge and citations. For this purpose, we have selected the most recent studies period from 2024 to 2025, and cited in the text. Please see the page 1 and 4, line#24, 30, 135-141, 161-166, 245-248, 886-910, highlighted by the YELLOW colour in the draft (section 1, section 2). However, there are still limited studies in the year 2025. Therefore, some studies e.g., Makun & Singh (2025) and Bajaj et al., (2025) substantially contributed to the literature by concentrating the impact of trade agreements and climate change on bilateral exports. Likewise, Khan et al., (2024) and Bing et al., (2024) also critically analysed the impact of trade agreements on bilateral exports. Nevertheless, several recent studies e.g., Tian et al., (2024), Liu et al., (2024), Thi Quy et al., (2024), Yasar (2024) and Wang et al., (2024) also emphasized on factor affecting such as geographic (transportation costs, spatio-temporal heterogeneity, complementarily and comparative advantage), climate change and trade agreements (trade preferential agreements), which are followings:

• Makun, K., & Singh, B. (2025). Trade deregulation and fiscal revenue in selected Pacific Island countries. PloS one, 20(1), e0315733.

• Bajaj, K., Mehrabi, Z., Kastner, T., Jägermeyr, J., Müller, C., Schwarzmüller, F., ... & Ramankutty, N. (2025). Current food trade helps mitigate future climate change impacts in lower-income nations. PloS one, 20(1), e0314722.

• Khan, H., Chen, Y., & Lv, L. (2024). Does the China–Pakistan free trade agreement benefit the vegetable exports of Pakistan? A gravity estimation. Frontiers in Sustainable Food Systems, 8, 1362910.

• Bing, Z. U. O., Gang, W. U., Chen ZHANG, S. N., Runji, L. I. N., Junyao, Y. I. N., Haowei, D. A. I., & Jiao, W. U. (2024). Research on the impacts of trade agreements on global trade flows: Based on complex network perspective. World Regional Studies, 33(1), 1.

• Tian, J., Zhu, Y., Hoang, T. B. N., & Edjah, B. K. T. (2024). Analysis of the competitiveness and complementarity of China-Vietnam bilateral agricultural commodity trade. Plos one, 19(4), e0302630.

• Liu, N., Li, Y., Jiang, M., & Liu, B. (2024). Trade shocks and trade diversion due to epidemic diseases: Evidence from 110 countries. Plos one, 19(5), e0301828.

• Thi Quy, N., Hai, N. C., & Dao, H. T. T. (2024). Time-varying causality relationships between trade openness, technological innovation, industrialization, financial development, and carbon emissions in Thailand. Plos one, 19(5), e0304830.

• Yaşar, T. (2024). Impact of Countries’ Logistics Performance on Their Exports: The Case of G-8 Countries. Journal of Transportation and Logistics, 9(1), 60-67.

• Wang, H., Wu, Y., & Zhu, N. (2024). Spatio-temporal heterogeneity of China’s import and export trade, factors influencing it, and its implications for developing countries’ trade. Plos one, 19(4), e0300307.

Reviewer #2: Recommendations:

Based on the findings and conclusions of the study, the following minor recommendations can be made:

Comment:

1.Expanding the Scope of the Analysis:

The authors acknowledge the limitations of the study and suggest that future research should incorporate additional factors, such as the impact of agglomeration, market access, and production costs, on bilateral exports (Section 6). Expanding the scope of the analysis can provide a more comprehensive understanding of the drivers of bilateral trade in Asian countries.

Response:

Thank you for your excellent comment. We have substantially improved above suggestion by expanding the scope of analysis concern over agglomeration, market access and production costs in the section 6. Please see the page#19, line#686-693, highlighted text in GREEN in the text section 6.

Comment:

2.Incorporating Detailed Geographic Factors:

While the study examines the impact of various geographic factors, such as access to the ocean and common borders, the authors could consider incorporating more detailed geographic factors, such as the total land-use area by mountains, deserts, and seashores, to further enhance the understanding of their influence on bilateral exports (Section 6).

Response:

This is also nice suggestion. As per your suggestion, we have also incorporated the detailed geographic factors such as total land-use area by mountain, deserts and seashores for further understanding of influence on bilateral exports. So, please see the page#19, line #673-684, , highlighted text in GREEN colour in the section

Comment:

Overall, this is a well-designed and executed study that contributes to the understanding of the role of geographic factors, climate change, and trade agreements in determining bilateral exports in Asian countries. The authors have thoroughly addressed the research objectives and have provided a comprehensive analysis supported by appropriate empirical methods and robust findings.

Response:

Thank you for nice comment.

---

## [Editor Report · Decision Letter 1]

18 Feb 2025

Climate change, geography and trade agreements: a perspective of Asian bilateral trade

PONE-D-24-32100R1

Dear Dr. Hussain,

We’re pleased to inform you that your manuscript has been judged scientifically suitable for publication and will be formally accepted for publication once it meets all outstanding technical requirements.

Kind regards,

Asif Khan, PhD Law

Academic Editor

PLOS ONE
---

## [Editor Report · Acceptance letter]

PONE-D-24-32100R1

PLOS ONE

Dear Dr. Hussain,

I'm pleased to inform you that your manuscript has been deemed suitable for publication in PLOS ONE. Congratulations! Your manuscript is now being handed over to our production team.

Kind regards,

on behalf of

Dr. Asif Khan

Academic Editor

PLOS ONE